# Factors associated with Alzheimer's disease prevalence and mortality in Brazil—An ecological study

**Murilo Bastos**[1,2]*, **Michael Pereira da Silva**[3], **Danyele da Silva**[4], **Glauco Nonose Negrão**[5], **Graziele Schumanski**[6], **Weber Claudio Francisco Nunes da Silva**[5], **Juliana Sartori Bonini**[1,5]

**1** Laboratório de Neurociência e Comportamento, Universidade Estadual do Centro-Oeste, Guarapuava, PR, Brazil, **2** Cline Research Center, Curitiba, PR, Brazil, **3** Faculty of Medicine, Federal University of Rio Grande, Rio Grande, RS, Brazil, **4** Departamento de Enfermagem, Universidade Estadual do Centro-Oeste, Guarapuava, PR, Brazil, **5** Associação de Estudos, Pesquisa e Assistência as Pessoas com Doença de Alzheimer, Universidade Estadual do Centro-Oeste, Guarapuava, PR, Brazil, **6** Profnit, Universidade Estadual do Centro-Oeste, Guarapuava, PR, Brazil

* murilo_bastos@yahoo.com.br

**Data Availability Statement:** The data used in this study are public, made available by the country's Ministry of Health and by entities linked to research and the government. (https://cidades.ibge.gov.br/

## Abstract

A few epidemiological studies are evaluating the prevalence and mortality rates of Alzheimer's disease, with no one using a nationwide sample of Brazilian elderlies. This study aims to calculate the prevalence of Alzheimer's disease and investigate possible associations with sociodemographic and lifestyle factors and the presence of diseases non-communicable, and the prevalence and mortality for all Brazilian state capitals. This is an ecological design study made with secondary public data provided by the Ministry of Health. Prevalence rates were calculated based on the analysis of the dispensing of Alzheimer's disease-specific drugs. Correlation analyzes were performed between rates and factors, and a multiple linear regression analysis was used to analyze possible associations between variables, controlled for each other. AD prevalence was 313/100,000. Prevalence rates were positively associated with primary health care coverage factors and negatively associated with ultra-processed food consumption and physical activity levels. AD mortality was 98/100,000. Mortality rates were positively associated with the proportion of obese elderly and elderly living on up to half the minimum wage and were inversely associated with the proportion of elderly with diabetes factors. We found positive and negative associations of sociodemographic, behavioral and diabetes indicators with Alzheimer's disease prevalence and mortality, which provide data that can be investigated by studies with different designs.

## Introduction

Aging causes a wide range of molecular and cellular damage which leads to a gradual decrease in physical and mental capacity, an increased risk of disease, and, ultimately, death [1]. It is estimated that by 2050, 21.4% of the world's population will be over 60 years old [1]. The Brazilian Institute of Geography and Statistics estimates that in 2060, 25.5% of Brazilians will be

brasil/panorama) (https://www.saudeidoso.icict.fiocruz.br/novo2/td_munic_5.php) (https://datasus.saude.gov.br/mortalidade-desde-1996-pela-cid-10) (http://svs.aids.gov.br/download/Vigitel/).

**Funding:** The authors received no specific funding for this work.

**Competing interests:** NO authors have competing interests.

over 65 years of age, and life expectancy might increase approximately 3 years (75.7 years in 2016 to 78.6 in 2030) [2].

Dementia syndromes are neurodegenerative diseases characterized by irreversible destruction of neurons resulting in a progressive and disabling loss of specific functions of the nervous system [3]. Alzheimer's Disease (AD) is the primary and most prevalent dementia, accounting for about 60 to 70% of cases [4]. AD develops insidiously, continuously and slowly, having advanced age as its main risk factor [3]. AD is also related to other factors such as reduced brain cognitive reserve, reduced brain size, low educational level, low occupational complexity, low levels of physical activity throughout life [5, 6], and poor nutritional status [7].

However, even with the increase in the elderly population and, consequently, the possible increase in people with AD in Brazil, few epidemiological studies evaluated the AD prevalence and mortality rates using a nationwide sample of Brazilian elderlies. In fact, data regarding AD prevalence are restricted to specific states or cities [8–12] limiting the better understanding of how AD is prevalent and impact the mortality rates in elderly Brazilian. Furthermore, it also limits the knowledge of possible factors associated with the disease, such as sociodemographic, lifestyle, and the presence of other chronic diseases.

The Brazilian Unified Health System (UHS) offers specific drugs for AD treatment free of charge. Thus, AD prevalence might be estimated using data of the number of elderlies who took these drugs from UHS. Therefore, this study aims to verify the AD's prevalence and mortality rates in all Brazilian state capitals. We also aimed to verify the association of sociodemographic, lifestyle, and other non-communicable diseases (NCDs) with AD prevalence and mortality.

## Methods

### Design and estimates of prevalence and mortality

It is an ecological epidemiological study based on secondary data from 2019 provided by the Brazilian Ministry of Health and the Health Indicators and Monitoring System for Elderly Policies (SISAP IDOSO) of the Oswaldo Cruz Foundation [13].

The Brazilian Unified Health System (UHS) offers specific drugs for AD treatment free of charge through the diagnosis of probable AD according to the Clinical Protocol and Therapeutic Guidelines [14]. This document establishes the criteria for the diagnosis and treatment and is essential for the dispensing of medication. We calculated the AD prevalence using the number of elderlies who took specific drugs to treat AD from the Brazilian UHS and the total elderly population of each state capital analyzed. The prevalence was expressed in cases for each 100,000 people (total number of patients/total population X 100,000 people). The data was obtained from the SISAP IDOSO database [13]. The AD's mortality rate, expressed in deaths for each 100,000, was obtained from the Mortality Information System, accessed through the Ministry of Health database [15] (https://datasus.saude.gov.br/mortalidade-desde-1996-pela-cid-10), filtering mortality by AD (ICD G30) for each state capital for people aged over 60 years.

### Sociodemographic factors

The Human Development Index by cities (HDIm), Gini Index, per capita income, were obtained from the Brazilian Institute of Geography and Statistics [16] (IBGE) and corresponded to the most current data available since the last census was carried out in 2010.

The educational level was obtained from the VIGITEL Brazil survey in 2019 [17]. In addition, data on the percentage of elderly living on half (1/2) minimum wage and percentage of

elderly living on a quarter (1/4) of the minimum wage were taken from SISAP IDOSO [13] and have 2010 as the year of reference.

Regarding the data on access to food, the Brazilian Scale of Food Insecurity was chosen, applied in the 2017–2018 Family Budget Survey, and made available by the IBGE portal [18], which ranks based on the score of nine questions about access to food and classifies the household in food security (FS) and food insecurity (FI), the latter still stratified into mild, moderate, and severe. However, in this study, we only used the FI data in the associations.

The data of primary health care coverage (PHC) were taken from the e-manager portal of the Ministry of Health [19] (https://egestorab.saude.gov.br/index.xhtml), reference year was 2019, with PHC coverage defined as estimated population coverage in Primary Care, given by the percentage of the population covered by teams from the Family Health Strategy and by teams of equivalent traditional Primary Care, parameterized concerning the population estimate.

## Lifestyle factors

The Physical Activity Level (PAL), considered as the percentage of elderly who practiced at least 150 minutes of moderate to vigorous physical activity (PA) per week; Ultra-processed foods consumption, considered as a percentage of the elderly population that consumes five or more servings of ultra-processed foods per day; Consumption of protective foods, considered as a percentage of the elderly population that consumes five or more minimally or unprocessed foods per day; Overweight, considered as a percentage of the elderly population with a body mass index (BMI) between 25 and 29.9 kg/m2; Obesity; considered as BMI $\geq$ 30kg/m2, were obtained from the VIGITEL 2019 survey [17].

## Statistical analysis

The Shapiro-Wilk test verified data normality. Pearson's correlation analyzed the association between mortality rates and obesity, physical activity level, hypertension, the proportion of elderly living on half the minimum wage, consumption of protective foods, food insecurity, HDIm, and Gini index. Spearman's correlation test analyzed the association between the variables prevalence and women mortality rates, and the variables ultra-processed foods consumption, high educational level, and PHC.

Multiple linear regression analyzed the independent association of sociodemographic, lifestyle factors, and the presence of NCDs with AD prevalence and mortality rates. The variables of each model were included hierarchically and we dropped variables with p>0.20 using the backward procedure. The statistical significance level was set at p<0.05. All analyzes were performed using SPSS® version 24.0 statistical software.

## Results

The AD prevalence for all Brazilian capitals was 313/100,000 for overall, 340/100,000 among men and 240/100,000 among women. The city of Porto Alegre, Rio Grande do Sul showed the lowest AD prevalence for the overall (42/100,000) and each sex (Men: 33/100,000; Women: 47/100,000). Teresina (Piauí) showed the highest AD prevalence for the overall (902/100,000) and for each sex (Men: 747/100,000; Women: 1,000/100,000).

The AD mortality for all Brazilian capitals was 98/100,000 for overall, 73/100,000 among men and 108/100,000 among women. The city of Rio Branco (Acre) showed the lowest AD mortality for the overall analysis (27/100,000) and each sex (Men: 33/100,000; Women: 47/100,000). Vitória (Espírito Santo) showed the highest AD mortality for the overall (193/100,000) and each sex (Men: 163/100,000; Women: 244/100,000).

**Table 1. Correlation between prevalence and mortality of Alzheimer's disease in 27 Brazilian capitals and sociodemographic, lifestyle, and non-communicable diseases (NCDs) factors.**

| Variables | Prevalence | | | Mortality | | |
|---|---|---|---|---|---|---|
| | Overall | Men's | Women's | Overall | Men's | Women's |
| **Sociodemographic** | | | | | | |
| High educational level | 0,13 (p = 0,52) | 0,07 (p = 0,73) | 0,12 (p = 0,54) | **0,54* (p<0,01)** | **0,48* (p = 0,01)** | **0,63* (p<0,01)** |
| HDIm | 0,13 (p = 0,52) | 0,10 (p = 0,63) | 0,13 (p = 0,53) | **0,59* (p<0,01)** | **0,48* (p = 0,01)** | **0,60* (p<0,01)** |
| Gini Index | 0,11 (p = 0,58) | 0,07 (p = 0,72) | 0,09 (p = 0,66) | 0,08 (p = 0,68) | -0,22 (p = 0,91) | 0,03 (p = 0,89) |
| Per Capita Income | -0,04 (p = 0,83) | -0,13 (p = 0,54) | -0,04 (p = 0,83) | **0,48* (0,01)** | **0,43* (p = 0,03)** | **0,54* (p<0,01)** |
| Elderly people living on up to 1/2 minimum wage (%) | 0,08 (p = 0,70) | 0,08 (p = 0,67) | 0,01 (p = 0,97) | **-0,56* (p<0,01)** | **-0,45* (p = 0,02)** | **-0,68* (p<0,01)** |
| PHC | 0,22 (p = 0,27) | 0,17 (p = 0,41) | 0,24 (p = 0,22) | -0,14 (p = 0,5) | -0,18 (p = 0,37) | 0,08 (p = 0,70) |
| Food Insecurity | -0,07 (p = 0,71) | -0,00 (p = 0,98) | -0,07 (p = 0,70) | **-0,41* (p = 0,03)** | **-0,41* (p = 0,03)** | **-0,44* (p = 0,02)** |
| **Lifestyle** | | | | | | |
| Ultra-processed foods | -0,25 (p = 0,20) | -0,31 (p = 0,12) | -0,26 (p = 0,18) | 0,18 (p = 0,37) | 0,11 (p = 0,60) | 0,25 (p = 0,21) |
| Protective foods | -0,00 (= 0,99) | -0,06 (p = 0,72) | 0,01 (p = 0,97) | 0,37 (0,06) | 0,31 (p = 0,12) | 0,39 (p = 0,05) |
| Obesity | -0,29 (p = 0,14) | -0,28 (p = 0,16) | -0,26 (p = 0,18) | -0,24 (p = 0,23) | -0,27 (p = 0,26) | -0,06 (p = 0,78) |
| PAL | -0,08 (p = 0,68) | -0,18 (p = 0,38) | -0,07 (p = 0,72) | 0,08 (p = 0,70) | -0,09 (p = 0,66) | 0,00 (p = 0,99) |
| **NCDs** | | | | | | |
| Hypertension | -0,03 (p = 0,87) | -0,18 (p = 0,35) | -0,17 (p = 0,40) | -0,01 (p = 0,94) | -0,05 (p = 0,78) | -0,02 (p = 0,93) |
| Diabetes | 0,003 (p = 0,99) | -0,00 (p = 0,99) | 0,07 (p = 0,74) | 0,12 (p = 0,55) | -0,13 (p = 0,51) | 0,06 (p = 0,75) |

Source: Own authorship.

* Indicate statistically significant values (p≤0.05). High educational level: education range between 15 and 20 years of study; HDIm: Human Development Index by cities; Elderly people with ½ minimum wage: Proportion of elderly people who receive up to half the minimum wage; PHC coverage: coverage of primary health care; Food Insecurity: Proportion of population with food insecurity according to the Brazilian Food Insecurity Scale; Ultra-processed foods: Daily consumption of at least 5 ultra-processed foods; Protective Foods: Daily consumption of at least 5 minimally/unprocessed foods. PAL: Physical activity level considering 3 domains.

Table 1 shows the correlation between sociodemographic, lifestyle, and NCDs factors with the AD's prevalence and mortality. We only found a significant correlation for mortality rates. The high educational level and HDI factors were positively correlated with AD mortality for the overall and each sex, with correlations ranging from r = 0.48 to 0.59. The proportion of elderly people living with up to half the minimum wage and food insecurity were inversely correlated with AD mortality for the overall and each sex (correlation ranging from r = -0.41 to -0.56).

A multiple linear regression analysis was used to verify factors independently associated with AD prevalence and mortality (Table 2). For AD prevalence, the regression analysis resulted in a statistically significant model ($F_{(5,20)}$ = 4,06 $R^2$ = 0,50, $R^2$ adjusted = 0,38, p = 0.01). The PHC coverage (B = 5.4, 95%CI = 1.15; 9.6, p = 0.01), consumption of ultra-processed foods (B = -41.8, 95%CI = -75.1; -8.5, p = 0.02) and physical activity level (B = -23.3, 95%CI = - 44.0; -2.6, p = 0.03) were significant predictors of AD prevalence.

The regression analysis for AD mortality rate also resulted in a statistically significant model ($F_{(3,22)}$ = 7,25; $R^2$ = 0,50; $R^2$ adjusted = 0,49, p = 0.001). The proportion of elderly living with half the minimum wage (B = - 3.2, 95%CI = -4.7; -1.7, p = 0.00), obesity (B = -4.1, 95%CI = -8.0; -0.3, p = 0.04) and diabetes prevalence (B = 4.9, 95%CI = 0.2; 9.6, p = 0.04) were predictors of AD mortality rates.

## Discussion

This study verified the prevalence and mortality rates of AD among elderly people of all Brazilian state capitals in 2019. In addition, the study found significant associations between

**Table 2. Multiple linear regression for factors associated with AD's prevalence and mortality rates for Brazilian capitals (n = 27).**

| Variables | Prevalence | | Mortality | |
|---|---|---|---|---|
| | Adjusted $R^2$ = 0,38 (Final Model) | | Adjusted R2 = 0,43 (Final Model) | |
| | B (95%CI) | p | B (95%CI) | p |
| High educational level | - | - | - | - |
| HDIm | - | - | - | - |
| Gini Index | - | - | - | - |
| Per Capita Income | - | - | - | - |
| Elderly people living on up to 1/2 minimum wage (%) | - | - | -3,2 (-4,7; -1,7) | <0,001 |
| PHC | 5,4 (1,15; 9,6) | 0,01 | - | - |
| Food Insecurity | - | - | - | - |
| Ultra-processed foods | -41,8 (-75,1; -8,5) | 0,02 | - | - |
| Protective foods | - | - | - | - |
| Obesity | - | - | -4,1 (-8,0; -0,3) | 0,04 |
| PAL | -23,3 (-44; -2,6) | 0,03 | - | - |
| Hypertension | - | - | - | - |
| Diabetes | - | - | 4,9 (0,2; 9,6) | 0,04 |

Source: Own authorship.—indicate variables with p>0.15 that did not enter the final regression model. B: linear regression coefficient; 95%CI: Confidence Interval; p: level of statistical significance; High educational level: education range between 15 and 20 years of study; HDIm; Human Development Index by cities; PHC coverage: primary health care coverage; Food Insecurity: Proportion of population with food insecurity according to the Brazilian Food Insecurity Scale. PAL: Physical activity level considering 3 domains.

prevalence rates with PHC coverage, ultra-processed foods consumption, and physical activity level, and between AD mortality rate and factors of the proportion of elderly who live on half the minimum wage, obesity, and diabetes.

Studies have shown variability in all type dementia's prevalence, with an estimated worldwide prevalence of 800/100,000 in persons 55 to 65 years old; 2670/100,000 in 65 to 74 years old and 5140/100,000 in 70 to 79 years old, and AD's prevalence of 3070/100,000 (3,1%) in persons with 60 to 69 years old; 108,800/100,000 (11%) in persons with 70 to 79 years old and 26150/100,000 (26%) in persons with age ≥ 90 years old [20]. However, we have dementia's prevalence rates ranging from 4.66% in Europe to 8.48% in Latin America, and still with significant variability within continents, with 4.66% in southern Europe, up to 6.88% in the north, and in southern Latin America, ranging from 5.7% in Venezuela to 8.3% in Argentina [21–26]. The AD's prevalence found in this study was 313/100,000 (0.31%), lower than the rates found in other countries, especially of South American countries. However, considering the difficulty in systematizing information regarding new AD cases in Brazil, we assume that cases may be underreported, a situation that already occurs about dementia, both in high-income countries [27] and in low-income countries [28, 29] Another aspect that may be related to underreporting of AD is the low adherence to treatment with anticholinesterase inhibitors since its side effects can be hardly tolerated resulting in the treatment withdrawal [30].

AD's mortality rates among all Brazilian capitals found in this study was 36 deaths per 100,000 elderlies. In 2019, dementia, including AD, was the 7th major cause of death worldwide [4] and, in Brazil, occupying 9th place in the ranking of elderly deaths in 2019 [15]. If we consider only high-income countries, AD mortality is the 2nd major cause of deaths [4]. Globally, AD and other dementias caused 1.9 million deaths in 2015 [31]. However, the true impact

of dementia and AD on the mortality rates might be underestimated [32] since AD can make the elderly more vulnerable to other diseases that lead to death [33], in addition to the difficulty in determining the number of deaths caused by AD due to the way of recording the cause of death [34]. Therefore, even if AD does not occupy a high place in the ranking of causes of death in Brazilian elderlies, AD may be associated with other comorbidities that lead to elderly deaths, reinforcing the need of further investigation.

This study found that capitals with higher PHC coverage had higher AD prevalence. PHC is characterized by a set of individual and collective health actions, covering aspects of health promotion, disease prevention, diagnosis, treatment, rehabilitation, harm reduction, and health maintenance [35]. Elderlies with the subjective cognitive decline with a higher number of medical appointments had more frequent memory complaints than those with a lower number of appointments [36]. In Brazil, the elderly with AD visits a clinician more frequently compared to the elderly without AD, on the other hand, these individuals had fewer appointments with specialists [37]. Brazilian state capitals with higher PHC coverage might be more successful in screening AD cases because the healthcare system is more accessible to the community, especially those with lowest economic level.

PAL prevalence was inversely related to AD's prevalence. For elderlies, the PA recommendation for substantial health benefits, including mental health, is 150–300 minutes of moderate-intensity, or 75–150 minutes of vigorous intensity, per week [38]. Physical inactivity presents a 12.7% population-attributable risk for AD worldwide and 21% for the US population [39]. In Brazil, the proportion of physically inactive elderly with AD was also higher compared to the elderly without AD [37]. Furthermore, a 20% reduction in the risk of developing dementia was attributed to engaging in any physical activity [36]. Therefore, our hypothesis was confirmed, that AD maybe was also influenced by the adequate PAL, engaging in PA practice can slow down cognitive decline in elderly people with high concentrations of tau protein in the brain compared to elderly people with low PAL [40].

A higher proportion of elderly who consume five or more servings of ultra-processed foods were inversely associated with the AD's prevalence. The western standard of diet, composed of a good amount of processed foods with high levels of saturated fat and sugar, results in considerable adverse health effects [41], and these effects are not different to dementia and AD. Diets with high amounts of saturated fat and sugar worsened Aβ-42 concentrations in cerebrospinal fluid (FCS), which is an AD marker, compared to low amounts of fat and sugar diets [42]. Furthermore, dietary patterns such as the Mediterranean Diet, characterized by high consumption of fruits, vegetables, vegetables and cereals, olive oil, and moderate consumption of alcohol and red meat, are associated with lower rates of cognitive decline and significant reductions in AD rates [43]. However, we found an inverse association between consumption of ultra-processed foods and the prevalence of AD. As both variables refer to the same year, maybe the time of exposure to the risk factor consumption of ultra-processed food consumption has not yet been sufficient to cause an impact on the Alzheimer's disease prevalence. Thus, further investigation should monitor the impact of ultra-processed food on the Alzheimer's disease using a longitudinal design.

Our study showed that state capitals with higher diabetes prevalence had higher AD mortality rates. Type 2 Diabetes Mellitus (T2D) increases the risk of several types of dementia, including AD, through the formation of vascular lesions and ischemia, changing metabolic processes in the central nervous system, maintaining a chronic inflammatory process [44]. Furthermore, due to numerous similarities between diabetes and AD, some authors have called AD "Type 3 Diabetes" [45]. However, there seem to be no significant differences in in-hospital mortality between AD patients who had or did not have T2D, with only the risk ratio for the sum of comorbidities being significant [46]. Therefore, as the present study found that capitals with a

higher prevalence of T2D have higher AD's mortality and judging by the fewer studies specifically evaluating this association, further investigation should be carried out to better understand this relationship. However, regarding the difference between the result found in this study and the literature, one aspect of the potential for metabolic overlap of the two diseases, which makes any conclusions more difficult [47].

Finally, our study found a significant and negative association between the obesity prevalence and the AD's mortality. Although the literature presents data related to a more accentuated cognitive impairment in obese individuals (BMI $\geq 30$ kg/m$^2$) [48, 49], the relationship with AD's mortality presents results conflicting. For example, a greater body mass index can be protective against AD's death and is inversely related to medial temporal atrophy [7]. On the other hand, lower BMI is associated with faster cognitive decline [50, 51]. These findings may seem somewhat contradictory, especially about a desirable BMI. Lower body mass may be associated with lower energy consumption and, consequently, nutrient intake below the physiological needs. However, there is also the assumption that a lower body mass is associated with a higher resting metabolic rate in patients with AD [52]. Therefore, this study indicated that a higher BMI can be protective against AD mortality which might be explained by a higher energy reserve. However further investigation on the individual level should explore how BMI mediates AD mortality.

The present study has limitations. The adopted ecological design uses as a sampling unit a group of individuals and non-individuals, and this can interfere with the results. However, the data analyzed here are representative of the population of the capitals and present essential data for future studies using different research designs. Another limitation is the method of estimating AD's prevalence that used data on free dispensing of specific drugs for AD treatment by the public health system, which would limit the sample to only users of this system. However, we believe that this limitation does not reduce the importance of our findings since the system's user population is quite large and representative.

In conclusion, this study found a low prevalence of AD among the Brazilian elderly (300 cases per 100,000 elderly). Furthermore, we found an AD's mortality rate of 36 deaths per 100,00 elderlies. AD's prevalence was positively associated with PHC coverage and negatively associated with ultra-processed foods consumption and PAL. AD's mortality rate was inversely associated with the proportion of obese elderly and elderly living on up to half the minimum wage and positively associated with diabetes prevalence.

## Acknowledgments

I am grateful for the collaboration of MPS, for data analysis and writing, GNN, for database suggestions, JSB, for the thorough review, and other authors for writing and final review.

## Author Contributions

**Conceptualization:** Murilo Bastos, Michael Pereira da Silva, Glauco Nonose Negrão, Graziele Schumanski, Weber Claudio Francisco Nunes da Silva, Juliana Sartori Bonini.

**Data curation:** Murilo Bastos, Michael Pereira da Silva.

**Investigation:** Murilo Bastos.

**Methodology:** Murilo Bastos, Michael Pereira da Silva, Danyele da Silva, Glauco Nonose Negrão, Juliana Sartori Bonini.

**Supervision:** Weber Claudio Francisco Nunes da Silva, Juliana Sartori Bonini.

**Validation:** Juliana Sartori Bonini.

**Writing – original draft:** Murilo Bastos, Michael Pereira da Silva, Danyele da Silva.

**Writing – review & editing:** Danyele da Silva, Glauco Nonose Negrão, Graziele Schumanski, Weber Claudio Francisco Nunes da Silva, Juliana Sartori Bonini.

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
