## [Decision Letter · Decision Letter 0]

21 Mar 2023

Factors associated with Alzheimer's disease prevalence and mortality in Brazil - An ecological study

PONE-D-22-16449

Dear Dr. Murilo Bastos,

We’re pleased to inform you that your manuscript has been judged scientifically suitable for publication and will be formally accepted for publication once it meets all outstanding technical requirements.

Kind regards,

Leonardo Costa Pereira, Doctor

Academic Editor

PLOS ONE

Journal Requirements:

1. Thank you for stating the following financial disclosure:

“There is no fund to this submission.”

e)        Please provide an amended Funding Statement that declares *all* the funding or sources of support received during this specific study (whether external or internal to your organization) as detailed online in our guide for authors at http://journals.plos.org/plosone/s/submit-now. 

f)        Please state what role the funders took in the study.  If any authors received a salary from any of your funders, please state which authors and which funder. If the funders had no role, please state: "The funders had no role in study design, data collection and analysis, decision to publish, or preparation of the manuscript."

Please send your amended statements by return email; we will change the online submission form on your behalf.

2. We note that Figures 1 & 2 in your submission contain [map/satellite] images which may be copyrighted. All PLOS content is published under the Creative Commons Attribution License (CC BY 4.0), which means that the manuscript, images, and Supporting Information files will be freely available online, and any third party is permitted to access, download, copy, distribute, and use these materials in any way, even commercially, with proper attribution. For these reasons, we cannot publish previously copyrighted maps or satellite images created using proprietary data, such as Google software (Google Maps, Street View, and Earth). For more information, see our copyright guidelines: http://journals.plos.org/plosone/s/licenses-and-copyright.

a. You may seek permission from the original copyright holder of Figures 1 & 2 to publish the content specifically under the CC BY 4.0 license. 

Reviewers' comments:

Reviewer's Responses to Questions

**Comments to the Author**

1. Is the manuscript technically sound, and do the data support the conclusions?

Reviewer #1: Yes

Reviewer #2: Yes

2. Has the statistical analysis been performed appropriately and rigorously? 

Reviewer #1: Yes

Reviewer #2: Yes

3. Have the authors made all data underlying the findings in their manuscript fully available?

Reviewer #1: Yes

Reviewer #2: Yes

4. Is the manuscript presented in an intelligible fashion and written in standard English?

Reviewer #1: Yes

Reviewer #2: Yes

5. Review Comments to the Author

Reviewer #1: The work is an ecological epidemiological study based on 2019 secondary data provided by the Brazilian Ministry of Health and the System of Indicators and Monitoring of Health Policies for the Elderly (SISAP IDOSO) of the Oswaldo Cruz Foundation.

The importance of the research and the analyses performed lies in the demonstration that the study verified the prevalence and mortality rates of Alzheimer's disease among the elderly in all Brazilian capitals in 2019.

In conclusion, this study found a low prevalence of Alzheimer's disease among the Brazilian elderly (300 cases per 100,000 elderly). In addition, they demonstrated an Alzheimer's disease mortality rate of 36 deaths per 100,00 elderly. The prevalence of Alzheimer's disease was positively associated with primary health care (PHC) coverage and negatively associated with primary health care (PHC) coverage. - The result may not be different for populations more broadly associated with consumption of ultra-processed foods and Physical Activity Level (PAL). Alzheimer's disease mortality rate was inversely associated with the proportion of obese elderly and elderly living on up to half the minimum wage and positively associated with diabetes prevalence.

Thus, I believe it is relevant to disclose the data and the analysis performed, since all the data presented and analyzed will favor future research regarding Alzheimer's disease.

Despite the potential of the information presented for future research, caution is recommended in the use of the data, since they come from a government source that requires further in-depth research in other databases. - The result may be different for broader populations.

Reviewer #2: The manuscript is well written, both in terms of aesthetics and spelling, making it easy to understand and read. A great positive point, for me, was the use of data provided by federal government agencies, emphasizing the importance of collecting this information not only for the population, but for the scientific community.

6. PLOS authors have the option to publish the peer review history of their article (what does this mean?). If published, this will include your full peer review and any attached files.

Reviewer #1: No

Reviewer #2: No

---

## [Editor Report · Acceptance letter]

8 May 2023

PONE-D-22-16449 

Factors associated with Alzheimer's disease prevalence and mortality in Brazil - An ecological study 

Dear Dr. Bastos:

I'm pleased to inform you that your manuscript has been deemed suitable for publication in PLOS ONE. Congratulations! Your manuscript is now with our production department. 

Kind regards, 

on behalf of

Dr. Leonardo Costa Pereira 

Academic Editor

PLOS ONE